# Defect induced improved capacitive performance of MnS incorporated MoO$_3$ nanocomposite for supercapacitor electrodes in aqueous electrolytes

Mizanur Rahaman[1,2*], Mehedi Hasan Prince[3], Saif Mahmud Bijoy[4], Zakaria Siddiquee[2], Muhammad Rakibul Islam[1*]

1 Department of Physics, Bangladesh University of Engineering and Technology, Dhaka, Bangladesh, 2 Department of Physics, Kent State University, Kent, Ohio, United States of America, 3 Department of Materials and Metallurgical Engineering, Bangladesh University of Engineering and Technology, Dhaka, Bangladesh, 4 Advanced Materials and Liquid Crystal Institute, Kent State University, Kent, Ohio, United States of America

* mrahaman@kent.edu (MR); rakibul@phy.but.ac.bd (RI)

## Abstract

Electrode materials play a crucial role in improving supercapacitor performance. In this work, MnS nanoparticles were incorporated into MoO$_3$ to form a MoO$_3$/MnS nanocomposite via hydrothermal synthesis, and the capacitive performance of the resulting supercapacitor electrodes was evaluated. Their electrochemical performances were studied in conjunction with KCl and Na$_2$SO$_4$ electrolytes. The generation of MoO$_3$/MnS nanocomposite was confirmed by XRD analysis and HR-TEM imaging. It is found that the MnS nanoparticles altered the morphology of MoO$_3$ from nanobelts to nanofibers and produced a defective, rough surface. The defective surface expanded the interlayer distance from 0.396 nm to 0.421 nm. In both ionic electrolytes, the MoO$_3$/MnS composite demonstrated higher capacitive performance than the pristine MoO$_3$. At 0.3 A g$^{-1}$ current density, the estimated specific capacitance of MoO$_3$/MnS was 387 F g$^{-1}$ and 335 F g$^{-1}$ in KCl and Na$_2$SO$_4$ electrolytes, respectively. In the symmetric two-electrode system, the MoO$_3$/MnS shows a specific capacitance of 297 F g$^{-1}$ at 1 A g$^{-1}$, with an energy density of 33.37 Wh kg$^{-1}$ and a power density of 450 W kg$^{-1}$. The MoO$_3$/MnS nanocomposite provides excellent 90% retention after 1000 continuous charging-discharging cyclic. The enhancement of electrochemical performance is attributed to the large surface area, defective morphology, and broader interlayer distance. This system bridges the gap between traditional batteries and capacitors, offering a unique approach to producing supercapacitor electrodes.

## 1 Introduction

The rapid advancement of human civilization and technology in recent times has been largely driven by the widespread use of fossil fuels. The burning of fossil

**Data availability statement:** All relevant data are within the manuscript and its Supporting Information files.

**Funding:** The authors (MR and MRI) gratefully acknowledge the financial support from the Ministry of Education, Government of Bangladesh, under grant 37.20.0000.004.33.020.23(Part-5). MRI also gratefully acknowledges the financial support from the Basic Research Grant provided by the Bangladesh University of Engineering and Technology No. songtha/R-60/Re-6714(06.03.2024). The funders had no role in study design, data collection and analysis, decision to publish, or preparation of the manuscript.

**Competing interests:** no competing interest.

fuels harms our environment and is being depleted rapidly. The use of sustainable energy, such as sunlight and wind, requires reliable energy storage. Electrical energy can be stored either electrochemically in batteries or electrostatically in capacitors. Batteries have ~50–200 Wh/kg energy density and low (~1–1000 W/kg) power density, while electrostatic capacitors have energy densities less than 0.1Wh/kg and power densities over 5000 W/kg [1]. The gap between batteries and capacitors has been partially bridged by supercapacitors, which are currently used in power conditioning and electric transportation. Supercapacitors offer notable advantages such as long cyclic life, relatively fast charge and discharge, and high-power density, which makes a bridge between batteries and capacitors [2,3]. Generally, electrode materials play a significant role in enhancing the performance of supercapacitors [4–7].

Transition-metal oxides such as molybdenum trioxide ($MoO_3$) have gained significant interest as electrodes due to their high theoretical capacitance (1,005 C/g), exceptional cation accommodation efficiency, favorable charge transfer ability, and semiconducting properties [8–10]. $MoO_3$ has three different polymorphs. Among them, $\alpha$-$MoO_3$ is remarkably important due to its structural anisotropy and the alternating stack of $MoO_6$ octahedra double layer bound by the Van Der Waals force along the [001] direction [11]. However, $MoO_3$ is prone to structural instability, has poor electrical conductivity, and limited rate capacity that reduces the electrode's electrochemical properties for supercapacitor applications [12]. The electrochemical performance of $MoO_3$ can be improved by enhancing its electrical conductivity, surface area, and interlayer spacing.

The incorporation of nanostructured materials is an efficient approach for improving the capacitance of the oxide-based materials. The incorporation enhances surface area, improves conductivity, expands interlayer distance, and generates electrochemically active sites — all of which help achieve better capacitive performance. [4] Several studies have been performed on the supercapacitor applications of $MoO_3$-based nanocomposites. Zhou et al. fabricated an Ag-decorated $MoO_3$ nanocomposite for a supercapacitor electrode in a liquid phase method [13]. They achieved the highest specific capacitance of 225 F $g^{-1}$ and 71.1% cyclic stability with 8% Ag content in the Ag@$MoO_3$ nanocomposite. Capacitive performance and cyclic stability may not fully reflect the stability of a supercapacitor electrode. Imran et al. used the hydrothermal method to produce intertwined porous $MoO_3$–MWCNT nanocomposites [14]. The measured specific capacitance is 210 F $g^{-1}$at the lowest scan rate, 5 mV $s^{-1}$. At the lowest scan rate, ion diffusion is very slow, resulting in a longer time to complete the full cycle. Sadananda et al. grew ZnO nanoparticles on $MoO_3$ via the solid-state impregnation-calcination method [15]. This nanocomposite is combined with carbon black, yielding a specific capacitance of 280 F $g^{-1}$ at a current density of 1 A $g^{-1}$. In this work, the carbon black modifies the nanocomposites' original capacitance.

Recently, MnS nanoparticles gained considerable attention due to their high theoretical capacitance, strong redox reactions, charge transfer kinetics, and higher electronic conductivity ($3.2 \times 10^3$ S $cm^{-1}$) compared to their oxide counterparts [16].

Nanostructured MnS demonstrates high ionic penetration and intercalation–deintercalation properties, contributing to the electrochemical stability of supercapacitors. Moreover, when combined with other materials, MnS shows improved performance [17]. The polymorph structure of MnS plays a vital role in improving the electrochemical performance of $MoO_3$ [18]. Moreover, $MoO_3/MoS_2$ binary nanocomposites have shown significantly high specific capacitance and cycling stability [19]. Another recent study on $MoO_3/MoS_2$ nanocomposite electrode has demonstrated ultrahigh capacity and excellent rate performance, highlighting the advantages of such hybrid structures for pseudocapacitive energy storage [20]. Other research has also studied morphology and defect-controlled α- [20] MoO3 structures, showing that morphologies with higher surface area and defect sites can significantly improve charge storage capacity [21]. Beyond $MoO_3/MoS_2$ systems, various metal sulfide-modified $MoO_3$ composites, including bimetallic sulfide/oxide heterostructures, have been shown to enhance electrical conductivity and charge transport, providing a promising strategy to overcome the inherently low conductivity of pure $MoO_3$ [22]. Although reports on MoO3/MnS nanocomposites specifically for supercapacitors are limited, results from related metal sulfide and MoO3/metal sulfide systems suggest that forming heterostructures and introducing defects may enhance electrochemical performance. Several $MoO_3$-sulfide-based systems have been explored for electrochemical energy storage, reports on MnS-incorporated $MoO_3$ nanobelts synthesized via hydrothermal processing and systematic evaluations in different aqueous electrolytes are lacking. In particular, the role of MnS-induced defects in enhancing the capacitive performance of $MoO_3$ has not yet been comprehensively investigated.

In this study, we used a facile hydrothermal approach to synthesize $MoO_3$ nanobelts and $MoO_3/MnS$ nanocomposites and studied their structural, morphological, and electrochemical properties. The hydrothermal process was chosen because it produced approximately 85–90% of the final product relative to the precursor mass. Repeated batches prepared under identical conditions demonstrated consistent phase purity and electrochemical performance, indicating reliable batch-to-batch reproducibility. The incorporation of MnS drastically changed the morphology of the $MoO_3$ nanobelts to nanofibers, generating defects and pores. The defective porous morphology increases the specific surface area, number of active sites, thereby enhances the nanocomposite's electrochemical properties. The nanocomposite shows specific capacitances of 387 F g$^{-1}$ and 335 F g$^{-1}$ at current density 0.3 Ag$^{-1}$ in 0.5M KCl and 0.5M $Na_2SO_4$ electrolytes, respectively. These results demonstrate remarkable enhancement of the capacitive performance of $MoO_3/MnS$ nanocomposite thus opening a new way for improving supercapacitor electrodes.

## 2 Experimental section

### 2.1 Synthesis of $MoO_3$, MnS and $MoO_3/MnS$ nanocomposites

For the hydrothermal synthesis of $MoO_3$ nanobelts and MnS nanoparticles, the precursors, Sodium Molybdate ($Na_2MoO_4.2H_2O$) and Hydrochloric Acid (HCl), Manganese (II) chloride tetra-hydrate ($MnCl_2.4H_2O$), and Hydrazine hydrate ($N_2H_4$), were purchased from Merck, India.

Initially, 0.08 M $Na_2MoO_4.2H_2O$ was dissolved in 120 ml of DI water and stirred. After that, HCl (~3 ml) was added dropwise, the pH was maintained at 1, and the solution was stirred for 30 minutes to form a homogeneous solution. The mix was subsequently moved to a Teflon-lined autoclave, heated for 24 hours at 150 °C, and allowed to cool to room temperature. Following centrifugation, the yield was cleaned more than three times with ethanol and DI water to eliminate impurities. First, a specific amount of $MnCl_2 \cdot 4H_2O$ and $C_2H_5NS$ was combined in 120 mL of DI water, stirred for 1 hour. The past solution was merged with hydrazine hydrate ($N_2H_4$), and the mixture was agitated for 2 hours to form a homogeneous solution. The solution was then transferred to the autoclave and heated to 180 °C for 24 hours. The precipitate was washed several times with ethanol and DI water and dried at 60 °C for 3 h on a hot plate to obtain the MnS nanoparticles.

To prepare the MnS-incorporated $MoO_3$ nanocomposite, first, 5 wt% MnS nanoparticles were dissolved in 50 mL DI water and sonicated for 1 hour. Then it was mixed with 70 mL of a $MoO_3$ precursor solution and stirred for 30 minutes. The

solution was heated in the autoclave to 150° C for 24 hours, then cooled to room temperature. Finally, the resulting $MoO_3$/MnS nanocomposite was washed numerous times and dried at 60 °C.

## 2.2 Characterization

The surface morphology of $MoO_3$ and $MoO_3$/MnS was examined by a field emission scanning electron microscope (JSM 7600, JEOL) and a transmission electron microscope (JEM 2100 F, JEOL). For the structural analysis, X-ray diffraction (XRD) patterns of the samples were obtained by utilizing an X-ray diffractometer (PANalytical Empyrean) equipped with a Cu-Kα X-ray source (λCuKα = 1.54278 Å). The electrodes' electrochemical behavior was tested using a CS310 (Cortest, China) workstation in standard three-electrode configurations with a modified graphite working electrode, an Ag/AgCl reference electrode, and a $1\ cm \times 1\ cm$ platinum counter electrode plate. 0.5 M $Na_2SO_4$ and 0.5 M KCl aqueous ionic electrolytes were used in a $0.1V - 0.7\ V$ voltage window. For the working electrodes, polyvinyl alcohol ($C_2H_4O$) and dimethyl sulfoxide ($C_2H_6OS$) solvents were mixed with the electrode materials. Afterwards, these mixed solutions were put uniformly (0.3 mg active mass loading) via drop casting on the surface of modified graphite electrode and dried 30 min at 60 °C.

## 3 Results and discussion

### 3.1 Morphological analysis

Fig 1(a) shows the FE-SEM image of $MoO_3$ nanobelts formed after hydrothermal reaction with 200–300 nm diameters and 5–10 μm lengths. The inset of Fig 1 (a) shows FE-SEM image of MnS nanoparticles. The $MoO_3$ nanobelts aggregate

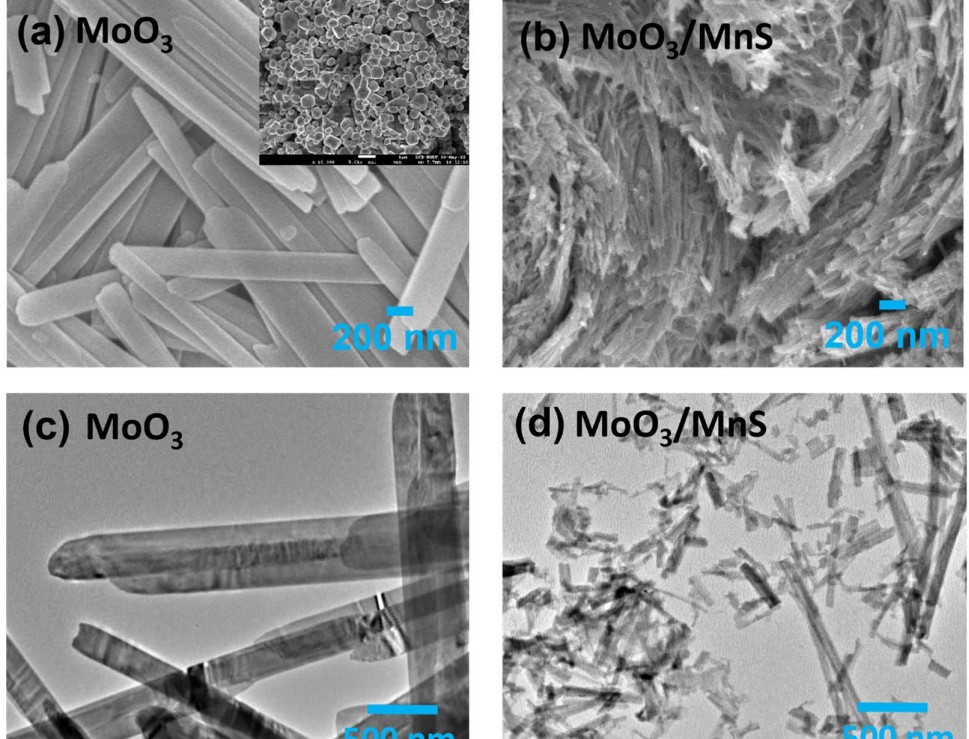

**Fig 1. Images of $MoO_3$ nanobelts and $MoO_3$/MnS nanocomposites. (a)** FE-SEM images of $MoO_3$ nanobelts; **(b)** FE-SEM images of $MoO_3$/MnS nanocomposites; **(c)** TEM images of $MoO_3$ nanobelts; **(d)** TEM images of $MoO_3$/MnS nanocomposites.

because of their large surface energy and surface tension, which minimizes their specific surface area [23]. After incorporation of MnS nanoparticles, the morphology of $MoO_3$ changed from nanobelts to nanofibers, as shown in Fig 1(b). By using ImageJ software, the measured nanofibers diameter is 30–40 nm. The MnS nanoparticles shrink to the width of the $MoO_3$ belts, resulting in an increase of specific surface area that improves the capacitive performance. The MnS nanoparticles are randomly distributed in the $MoO_3$ medium, creating a rough surface and numerous pores in $MoO_3$. Such rough and porous morphology of the $MoO_3$/MnS nanocomposites significantly increases the specific surface area, therefore, supplies more electrochemically active sites, which can accommodate the volume expansion [24]. This enhances the overall capacitive performance of MnS injected $MoO_3$ nanocomposites. Due to this unique morphology, the impedance of the material greatly reduces and provides rapid charge transportation, thereby enhancing capacitance [25].

Fig 1 (c, d) shows TEM micrographs of $MoO_3$ nanobelts and $MoO_3$/MnS nanocomposites. From TEM micrographs, it is evident that $MoO_3$ forms nanobelts that connect with one another. In the case of $MoO_3$/MnS nanocomposites, it is evident that the nanobelts break down, producing rich, porous nanofibers with defects.

Fig 2 demonstrates HR-TEM images of $MoO_3$ and the $MoO_3$/MnS nanocomposite, with clearly visible lattice fringes.

The irregular lattice fringes and distorted lattice planes suggest the presence of defect related structural distortion in the nanocomposite [26]. The measured interlayer spacing of $MoO_3$ nanobelts was 0.396 nm, which is consistent with previous experimental results. After the incorporation of MnS nanoparticles, the interlayer spacing of the nanocomposite has increased to 0.421 nm. The observed increase in interlayer spacing is primarily due to defect formation induced by MnS incorporation. In addition, local lattice strain and interfacial distortion at the $MoO_3$/MnS heterojunctions may also contribute to the lattice expansion. Such combined structural effects facilitate shortened diffusion pathways and may enhance ion accessibility within the electrode material [27]. The increased interlayer distance enhances the material's stability and reduces charge collapse between layers. Meanwhile, the pores of the nanocomposite accumulate more ions and potentially act as channels for prompt transportation [28]. While any spectroscopic techniques such as XPS, Raman, or EPR could enable quantitative analysis of defect states, the present conclusions are based on qualitative structural and diffraction evidence.

### 3.2 Structural analysis

The crystal structure of $MoO_3$ and $MoO_3$/MnS was investigated by X-ray diffraction (XRD), as shown in Fig 3.

The diffraction pattern of $MoO_3$ in Fig 3(b) displays sharp, well-defined peaks, which are characteristic of a highly crystalline material. The diffraction peaks at 12.7°, 23.3°, 25.7°, 27.4°, and 39.0° correspond to the (020), (110), (040),

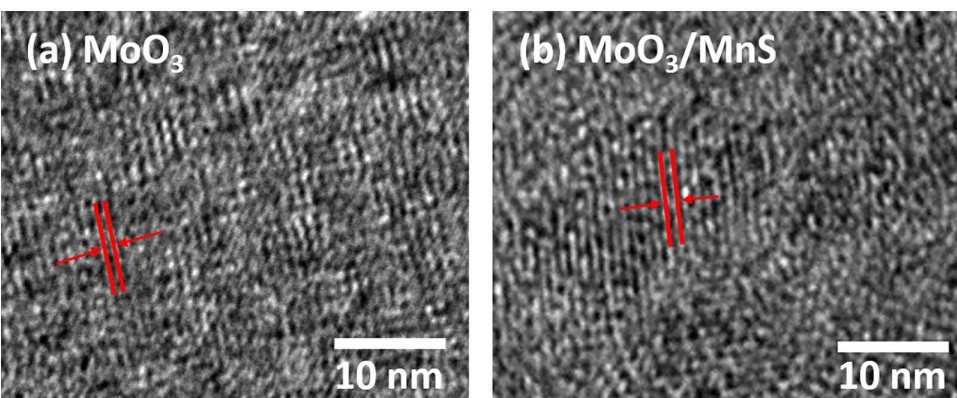

**Fig 2. HR-TEM images of (a) pristine $MoO_3$ nanobelts, (b) MnS-incorporated $MoO_3$ ($MoO_3$/MnS) nanocomposites.**

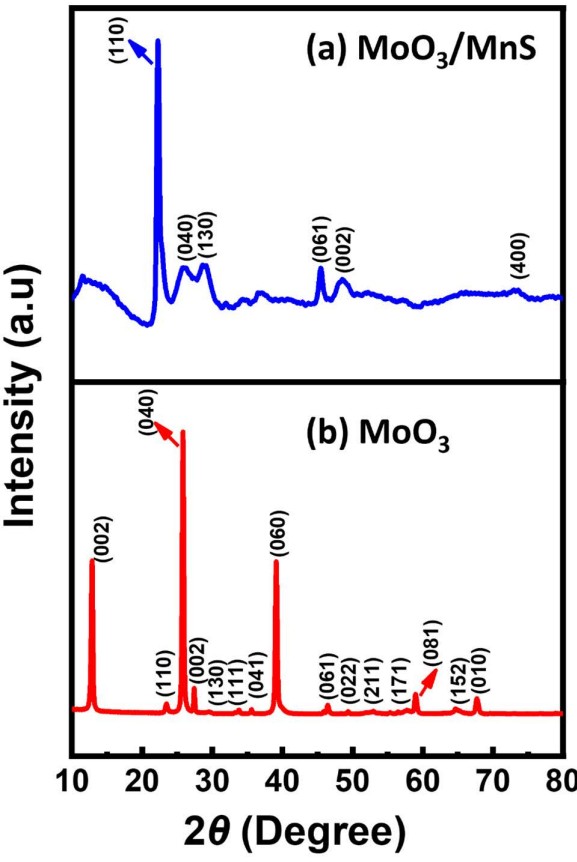

**Fig 3. X.ray diffraction pattern of (a) MoO₃/MnS nanocomposite and (b) MoO₃.**

(021), and (060) planes of orthorhombic (Pbnm space group) $MoO_3$ (JCPDS card No. 35–0609), respectively [29]. The prominent peak at 25.7° indicates anisotropic growth along the (004) plane during the synthesis process [30]. The sharp peaks suggest relatively large crystallite sizes. The diffraction peaks of MnS match the JCPDS card number 06–0518, corresponding to cubic α-MnS with a rock-salt structure (S1 Fig). The relatively weak MnS peaks may be attributed to their low weight fraction and uniform dispersion within the $MoO_3$ matrix. The XRD pattern of the $MoO_3$/MnS nanocomposite in Fig 3(a) shows notable differences. While some of the original peaks of $MoO_3$ are retained, additional peaks of MnS appear, confirming the successful formation of $MoO_3$/MnS nanocomposite. The characteristic peaks of MnS, at $2\theta \approx 29°$ and $2\theta \approx 49°$ can be assigned to the (111) and (220) cubic MnS, respectively. The coexistence of the $MoO_3$ and MnS peaks in the nanocomposite's XRD pattern indicates that the two phases remain separate and that no new phase is forming. In addition, some $MoO_3$ peaks are eliminated in the nanocomposite after incorporation of MnS, indicating increased disorder in $MoO_3$ [31]. MnS particles embedded within or at the grain boundaries of $MoO_3$ may introduce defects that reduce the material's overall crystalline, leading to missing XRD peaks [32]. The $MoO_3$ peaks in the nanocomposite's XRD pattern are broader and slightly shifted compared to those in the pure $MoO_3$ sample. The disappearance and broadening of certain $MoO_3$ diffraction peaks in the $MoO_3$/MnS nanocomposite may be attributed to reduced crystallite size, defects, and lattice distortion induced by MnS incorporation, as confirmed by HR-TEM (Fig 2). The Scherrer equation for crystallite

size estimation was not applied, as its underlying assumptions are not fully valid for this defective nanocomposite system. Moreover, the peak shift to higher angles may be attributed to the introduction of compressive strain in $MoO_3$.

Although direct elemental mapping by STEM-EDS was not performed in this study, the incorporation of MnS is supported by XRD, which is also consistent with the morphological transformation observed in TEM, and defect-induced lattice distortion in HR-TEM. Given the low MnS loading (5 wt%), MnS is expected to be finely dispersed within the $MoO_3$ matrix.

## 3.3 Electrochemical analysis

Cyclic voltammetry (CV) measurement was performed to investigate the charge storage mechanisms in the synthesized $MoO_3$ and $MoO_3$/MnS nanocomposite. The CV curves were recorded at a 40 mV s$^{-1}$ scan rate, as shown in Fig 4 (a-b). It is observed that the area enclosed within the CV curves is higher for $MoO_3$/MnS nanocomposite than pure $MoO_3$ for both 0.5 M $Na_2SO_4$, and 0.5 M KCl electrolytes. In case of the 0.5 M $Na_2SO_4$, the area for $MoO_3$ is $4.22 \times 10^{-4}\ A \bullet V \bullet cm^{-2}$, while the $MoO_3$/MnS nanocomposite exhibits a larger area of $4.58 \times 10^{-4}\ A \bullet V \bullet cm^{-2}$. Similarly, in 0.5 M KCl, the areas are $4.29 \times 10^{-4}\ A \bullet V \bullet cm^{-2}$ and $5.08 \times 10^{-4}\ A \bullet V \bullet cm^{-2}$ for $MoO_3$ and $MoO_3$/MnS, respectively. This increase in the enclosed area for the $MoO_3$/MnS nanocomposite indicates an enhancement in specific capacitance compared to pure

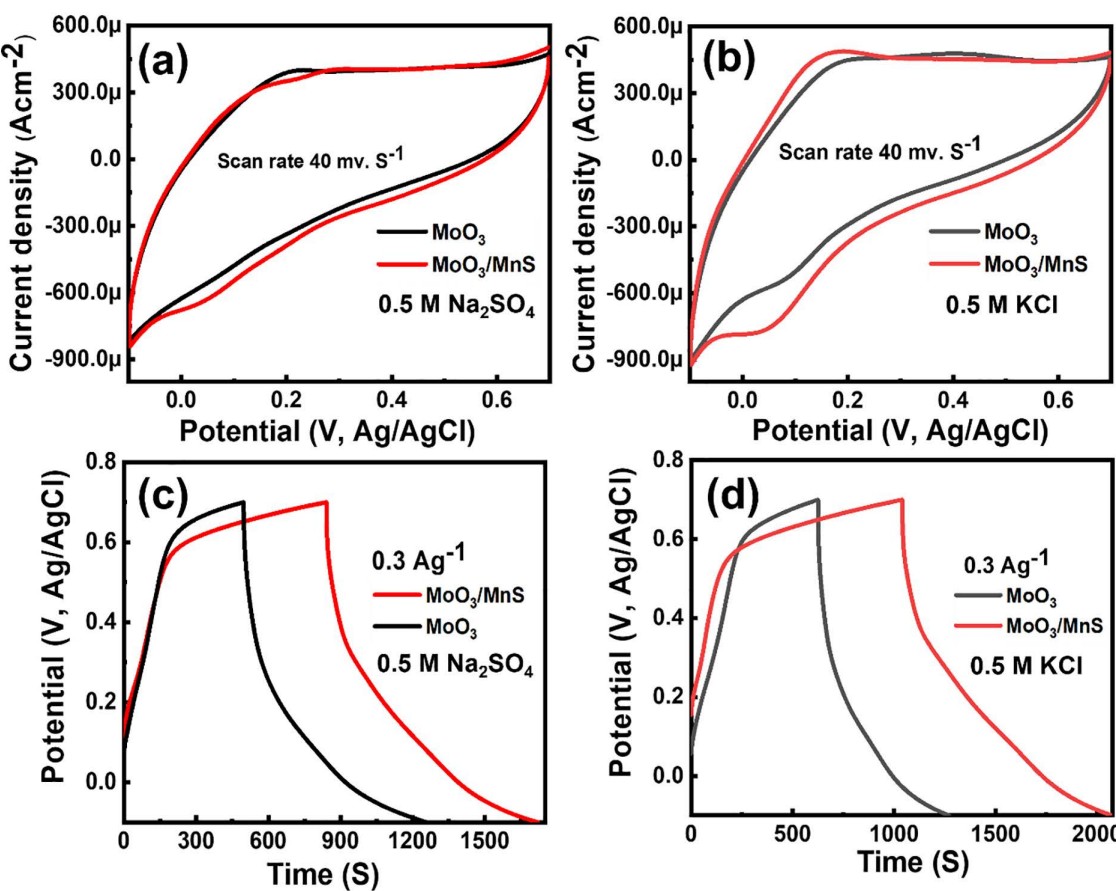

**Fig 4. CV curves at 40 mVs$^{-1}$ scan rate for $MoO_3$ (a) and $MoO_3$/MnS (b); Galvanostatic Charge Discharge (GCD) curves with constant current density 0.3 Ag$^{-1}$ in three three-electrode system of (c) $MoO_3$/MnS in $Na_2SO_4$, (d) $MoO_3$/MnS in KCl.**

$MoO_3$, signifying its superior charge storage capability [30]. The area under the CV curves, is noticeably greater when KCl is used as the electrolyte compared to $Na_2SO_4$. This may be due to the smaller ionic radius and higher mobility of $K^+$ ions in KCl, which enhance charge storage and ion transport within the electrode structure, as observed in previous studies [33].

For the $MoO_3$ sample, the CV curves in both electrolytes exhibit a quasi-rectangular shape, suggesting that its charge storage mechanism is influenced by a combination of double-layer capacitance (EDLC) and pseudo-capacitive processes [34]. In contrast, the $MoO_3$/MnS nanocomposite shows a more noticeable deviation from the rectangular shape, especially in the 0.5 M KCl electrolyte. From the CV analysis ($i = av^b$), the b value is 0.78 that indicates the contributions of both capacitive and diffusion controlled. This indicates the dominant contribution is faradaic pseudo-capacitive charge storage [5]. In the absorption/desorption reaction mechanism, alkali cations ($Na^+$ and $K^+$) are adsorbed and desorbed into the electrode/electrolyte interface, as illustrated below:

$$\left(MoO_3\right)_{surface} + M^+ + e^- \leftrightarrow \left(MoO_3 \bullet M\right)_{surface}$$

$$\left(MoO_3/MnS\right)_{surface} + M^+ + e^- \leftrightarrow \left\{\left(MoO_3/MnS\right) \bullet M\right\}_{surface}$$

Here, $M^+$ represents the $Na^+$ and $K^+$ present in two different electrolytes. In the intercalation mechanism, cations are intercalated into and then extracted from the electrode material. This process involves a reversible structural change, which contributes to the faradaic charge storage.

The intercalation mechanism can be represented as:

$$MoO_3 + M^+ + e^- \leftrightarrow \left\{\left(MoO_3\right) .M\right\}_{intercalation}$$

$$\left(MoO_3/MnS\right) + M^+ + e^- \leftrightarrow \left\{\left(MoO_3/MnS\right) \bullet M\right\}_{intercalation}$$

The improvement in the specific capacitance of the $MoO_3$/MnS nanocomposite can be attributed to several factors. The nanocomposite defective porous morphology with increased surface area, as evident from the SEM images Fig 1(a-b), may facilitate better electrolyte piercing and ion movement, thereby enhancing the charge storage capabilities [35]. Additionally, the incorporation of MnS not only improves electrical conductivity but also provides more active sites for redox reactions, which are crucial for pseudo-capacitance. However, MnS does not serve as the dominant pseudocapacitive phase; rather, it acts as a conductive, defect-inducing component that modifies the $MoO_3$ lattice, facilitating enhanced ion diffusion and charge transport.

Fig 4 (c-d) shows the galvanostatic charging curves of $MoO_3$ and $MoO_3$/MnS electrodes at different electrolytes at a constant 0.3 A$g^{-1}$ current density. The discharging time of the $MoO_3$/MnS is much longer than that of the $MoO_3$ electrode [15]. These indicate improvement in charge storage capacity, as shown in the CV curve. The specific capacitance (Cs) from GCD (Galvanostatic Charge Discharge) curves could be determined by applying the following formula $C_s = \frac{I \times \Delta t}{\Delta V \times m}$, where I = discharge current, $\Delta t$ = discharging time, $\Delta V$ = potential window, and m = deposited mass of working electrode. The specific capacitances of $MoO_3$ in $Na_2SO_4$ and KCl electrolytes are estimated to be 285 F g$^{-1}$ and 243 F g$^{-1}$, respectively. After inserting the MnS nanoparticles, the $MoO_3$/MnS electrode exhibits higher specific capacitance in both electrolytes, and the values are 335 F g$^{-1}$ in $Na_2SO_4$ and 387 F g$^{-1}$ in KCl. The specific capacitance is enhanced due to the porous morphology and the surface defects.

S2b Fig shows the GCD curve of the $MoO_3$/MnS electrode in the symmetric two-electrode system, where the specific capacitance is 297 F g$^{-1}$, the energy density is 33.37 W h kg$^{-1}$, and the power density is 450 W kg$^{-1}$. In addition,

the nanocomposite exhibits higher capacitance in KCl electrolyte than in $Na_2SO_4$ electrolyte. Generally, $K^+$ ions have higher molar conductivity than $Na^+$, which easily migrates into the electrode/electrolyte surface [36]. In case of $Na_2SO_4$, the $SO_4^{2-}$ anion will reduce the mobility of $Na^+$ cation, thus making the electrochemical process less effective. Additionally, the $K^+$ ions have a lower hydration radius that also helps to increase overall capacitance of the nanocomposite [37].

Electrochemical impedance spectroscopy is another measurement to characterize the electrode materials. From Fig 5(a) exhibits the Nyquist spectra of $MoO_3$ and $MoO_3$/MnS nanocomposite. In the Warburg region, the $MoO_3$/MnS curve is stepper than $MoO_3$ which indicates $MoO_3$/MnS has low Warburg resistance. The decreased Warburg resistance expedites ion transport and improves capacitive performance.

Cyclic stability is another critical measurement to characterize supercapacitors. Fig 5(b) shows the cyclic stability of $MoO_3$/MnS electrode for 4000 charging-discharging cycles. The $MoO_3$/MnS electrode exhibits 87% cyclic stability. The enhancement of the nanocomposite's cyclic stability suggests that the electrode shows a higher specific capacitance as observed for metal sulfide base nanocomposites [38].The common electrochemical activation energy in electrochemistry is the cause of this phenomenon and indicates excellent stability of the novel $MoO_3$/MnS.

To summarize, MnS is incorporated into $MoO_3$ nanobelts via a two-step hydrothermal method. The hydrothermal process employed in this study is facile and cost-effective. Its high yield and consistent batch reproducibility at the laboratory scale indicate significant potential for future scale-up and industrial application. The diffraction analysis confirmed the successful formation of $MoO_3$ nanobelts and $MoO_3$/MnS nanocomposite. The incorporated MnS nanoparticles produced defective porous nanofibers and expanded the inter layer spacing. These distinctive features allow ions to move promptly between their interfaces, providing effective charge storage. The $MoO_3$/MnS nanocomposite shows 387 F $g^{-1}$ and 335 $Fg^{-1}$ specific capacitances in KCl and $Na_2SO_4$ electrolytes, respectively. The power density of the nanocomposite is 450 W $kg^{-1}$ with an energy density of 33.37 W h $kg^{-1}$.The defective porous structure of $MoO_3$/MnS obtained by this simple technique provides an efficient way to produce high quality supercapacitor electrodes for energy storage applications.

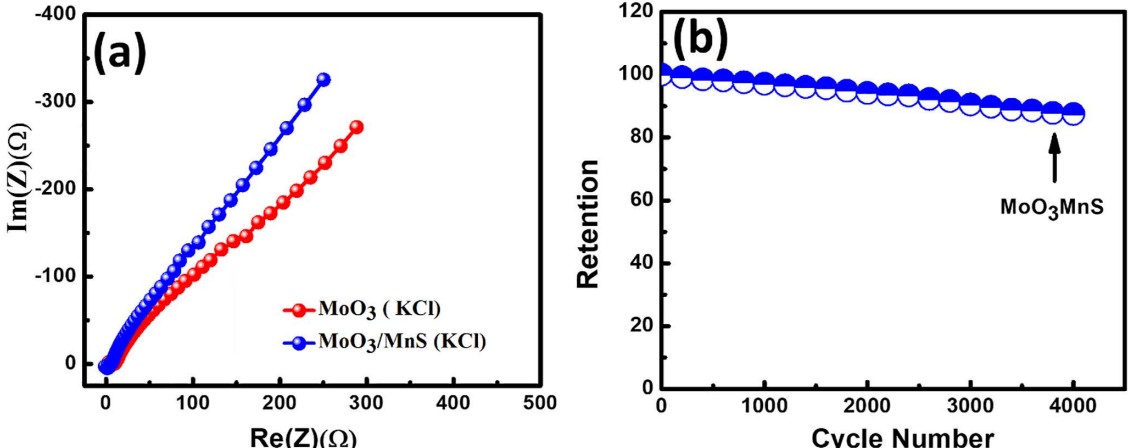

**Fig 5. Electrochemical impedance spectra of (a) $MoO_3$ and $MoO_3$/MnS and (b) Cyclic stability of $MoO_3$/MnS nanocomposite in KCl solution.**

## Supporting information

**S1 Fig. X-ray diffraction pattern of MnS nanoparticles.**
(DOCX)

**S2 Fig. CV curve of MoO$_3$/MnS (a), and GCD curve of MoO$_3$/MnS nanocomposite (b) at two electrode system.**
(DOCX)

**S1 File. All relevant data.**
(DOCX)

## Acknowledgments

The authors also acknowledge Professor Dr. Antal Jákli for correcting the manuscript and checking the validity of the data.

## Author contributions

**Conceptualization:** Zakaria Siddiquee.

**Data curation:** Mizanur Rahaman, Mehedi Hasan Prince, Saif Mahmud Bijoy.

**Formal analysis:** Mizanur Rahaman, Mehedi Hasan Prince, Saif Mahmud Bijoy, Muhammad Rakibul Islam.

**Funding acquisition:** Muhammad Rakibul Islam.

**Investigation:** Saif Mahmud Bijoy.

**Resources:** Zakaria Siddiquee.

**Supervision:** Muhammad Rakibul Islam.

**Validation:** Zakaria Siddiquee, Muhammad Rakibul Islam.

**Visualization:** Muhammad Rakibul Islam.

**Writing – original draft:** Mizanur Rahaman, Mehedi Hasan Prince, Muhammad Rakibul Islam.

**Writing – review & editing:** Mizanur Rahaman, Zakaria Siddiquee, Muhammad Rakibul Islam.

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
