## [Decision Letter · Decision Letter 0]

5 Jan 2026

PONE-D-25-65983Defect Induced Improved Capacitive Performance of MnS Incorporated MoO3 Nanocomposite for Supercapacitor Electrodes in Aqueous ElectrolytesPLOS One

Dear Dr.  Rahaman,

Thank you for submitting your manuscript to PLOS ONE. After careful consideration, we feel that it has merit but does not fully meet PLOS ONE’s publication criteria as it currently stands. Therefore, we invite you to submit a revised version of the manuscript that addresses the points raised during the review process.

Please submit your revised manuscript by Feb 19 2026 11:59PM. If you will need more time than this to complete your revisions, please reply to this message or contact the journal office at plosone@plos.org. Please include the following items when submitting your revised manuscript:

We look forward to receiving your revised manuscript.

Kind regards,

Latha Marasamy, Ph.D

Academic Editor

PLOS One

Journal Requirements:

Improved electrochemical performance of defect-induced supercapacitor electrodes based on MnS-incorporated MnO2 nanorods-10.1039/d4na00085d. eCollection 2024 Aug 6.

Antiferroelectric Bent-Core Liquid Crystal for Possible High-Power Capacitors and Electrocaloric Devices-10.3390/cryst10080652

In your revision ensure you cite all your sources (including your own works), and quote or rephrase any duplicated text outside the methods section. Further consideration is dependent on these concerns being addressed.

“Ministry of Education, Government of Bangladesh, under grant 37.20.0000.004.33.020.23(Part-5)

Bangladesh University of Engineering and Technology No. songtha/R-60/Re-6714(06.03.2024).”

6. Please note that funding information should not appear in any section or other areas of your manuscript. We will only publish funding information present in the Funding Statement section of the online submission form. Please remove any funding-related text from the manuscript.

7. In the online submission form, you indicated that “Data are available on request”

Additional Editor Comments:

Authors are suggested to revise the manuscript thoroughly by addressing each reviewer's comments.

Reviewer's Responses to Questions

**Comments to the Author**

1. Is the manuscript technically sound, and do the data support the conclusions?

Reviewer #1: Partly

Reviewer #2: Partly

2. Has the statistical analysis been performed appropriately and rigorously? 

Reviewer #1: Yes

Reviewer #2: No

3. Have the authors made all data underlying the findings in their manuscript fully available?

Reviewer #1: No

Reviewer #2: Yes

4. Is the manuscript presented in an intelligible fashion and written in standard English?

Reviewer #1: Yes

Reviewer #2: Yes

5. Review Comments to the Author

Reviewer #1: The manuscript entitled “Defect Induced Improved Capacitive Performance of MnS Incorporated MoO3 Nanocomposite for Supercapacitor Electrodes in Aqueous Electrolytes” reports the hydrothermal synthesis of a MoO3/MnS nanocomposite for supercapacitor applications and demonstrates enhanced electrochemical performance compared to pristine MoO3. While the study is relevant and the results are promising, the following minor issues should be addressed to improve clarity, completeness, and consistency.

1. Electrochemical impedance spectroscopy data are not provided, despite frequent claims of reduced impedance, improved conductivity, and enhanced ion transport. Inclusion of Nyquist plots and fitted equivalent circuit analysis would help support these claims and strengthen the electrochemical discussion.

2. The manuscript repeatedly attributes performance enhancement to increased surface area and porosity; however, no BET surface area measurements are reported. BET analysis is recommended to quantitatively support these claims.

3. The choice of KCl and Na2SO4 electrolytes should be more clearly justified, particularly with respect to the electrochemical stability window, ion size and hydration radius, and comparison with more commonly used electrolytes such as KOH or NaOH.

4. The cycling stability is evaluated for only 1000 charge–discharge cycles, which is relatively limited for supercapacitor studies. Recent literature commonly reports stability over 2000-10,000 cycles. The authors should either extend the cycling test or provide a justification for this limitation.

5. Several figure captions lack sufficient detail, such as magnification or scale bars. Figures should be consistently labeled (a, b, c, d) and clearly referenced in the text. Additionally, the phrase “the insect shows” in the SEM description appears to be a typographical error and should be corrected.

6. The manuscript shows inconsistencies in font size and formatting. Some figure captions appear in italics while others do not. The authors should carefully review the manuscript to ensure consistent formatting throughout.

Reviewer #2: 1. The manuscript discusses MnS incorporation effects extensively; however, no standalone FESEM image of pristine MnS nanoparticles is provided, while FESEM images are shown only for MoO₃ and MoO₃/MnS. This omission makes it difficult to confirm MnS size, dispersion, and morphology prior to composite formation.

2. The authors claim that MnS “shrinks the MoO₃ nanobelts into nanofibers,” yet no quantitative statistical analysis (diameter distribution, fiber length histograms) is provided to support this transformation.

3. The successful incorporation and spatial distribution of MnS within the MoO₃ matrix are not convincingly demonstrated. STEM-EDS elemental mapping is essential to confirm homogeneous Mn and S distribution.

4. Defects are inferred primarily from HRTEM lattice distortion. However, no complementary techniques such as XPS (Mo⁶⁺/Mo⁵⁺ ratio), Raman defect analysis, or EPR are used to quantitatively validate defect density.

5. The reported increase from 0.396 nm to 0.421 nm is attributed to defect generation, but alternative causes such as strain, intercalation, or phase distortion are not discussed or ruled out.

6. MnS polymorphism strongly influences electrochemical behavior, yet the manuscript does not clarify which MnS phase (α, β, γ) is present, nor its electrochemical relevance.

7. The authors attribute missing MoO₃ peaks to disorder induced by MnS incorporation, but crystallite size estimation (Scherrer analysis) or Williamson–Hall strain analysis is absent.

8. The enhancement in capacitance is repeatedly attributed to increased surface area, but no BET or pore-size distribution data are provided to support this central claim.

9. The CV analysis lacks b-value determination (i = aνᵇ) to differentiate capacitive vs diffusion-controlled contributions, which is critical for pseudocapacitive systems.

10. Claims of reduced impedance and improved charge transport are made without Nyquist plots or fitted equivalent circuit models.

11. The manuscript does not clearly report active material mass loading per electrode, which is essential for benchmarking specific capacitance values.

12. Stability is evaluated only up to 1000 cycles, which is significantly lower than contemporary supercapacitor standards (≥5000–10000 cycles).

13. The symmetric device performance is reported, but no Ragone plot comparison with recent MoO₃-based devices is provided.

14. The explanation for superior KCl performance is qualitative. Ion diffusion coefficients or EIS-based electrolyte resistance comparisons are not presented.

15. The Introduction lacks a systematic comparison with recent MoO₃/MnS, MoO₃/MoS₂, MoO₃/metal sulfide, and defect-engineered MoO₃ composites reported in the last 5 years.

16. A benchmark table comparing specific capacitance, energy density, cycling life, and synthesis method with similar MoO₃-based composites is missing.

17. The statement “no prior work has been done” is too strong. Several MoO₃–sulfide hybrid systems exist, and the novelty should be reframed more cautiously.

18. The manuscript does not separate MnS redox contribution vs MoO₃ intercalation contribution, nor does it provide supporting redox peak analysis.

19. Although hydrothermal synthesis is claimed to be scalable, no yield, batch reproducibility, or scale-up discussion is included.

20. Several sections contain grammatical errors, repetitive explanations, and unclear phrasing that obscure scientific meaning, requiring professional language editing.

6. PLOS authors have the option to publish the peer review history of their article (what does this mean?). If published, this will include your full peer review and any attached files.

Reviewer #1: No

Reviewer #2: **Yes:** RAGURAM THANGAVEL

---

## [Author Response · Author response to Decision Letter 1]

27 Jan 2026

I am grateful for reviewer and editor comments. I tried to answer all the comments and include valuable information. Some of the questions of the second reviewer were very fantastic.

---

## [Decision Letter · Decision Letter 1]

23 Apr 2026

Defect Induced Improved Capacitive Performance of MnS Incorporated MoO3 Nanocomposite for Supercapacitor Electrodes in Aqueous Electrolytes

PONE-D-25-65983R1

Dear Dr. Mizanur Rahaman,

We’re pleased to inform you that your manuscript has been judged scientifically suitable for publication and will be formally accepted for publication once it meets all outstanding technical requirements.

Kind regards,

Latha Marasamy, Ph.D

Academic Editor

PLOS One

Additional Editor Comments (optional):

The authors have significantly enhanced the manuscript's quality. Best Wishes.

Reviewers' comments:

Reviewer's Responses to Questions

**Comments to the Author**

1. If the authors have adequately addressed your comments raised in a previous round of review and you feel that this manuscript is now acceptable for publication, you may indicate that here to bypass the “Comments to the Author” section, enter your conflict of interest statement in the “Confidential to Editor” section, and submit your "Accept" recommendation.

Reviewer #1: (No Response)

Reviewer #3: All comments have been addressed

2. Is the manuscript technically sound, and do the data support the conclusions?

Reviewer #1: Partly

Reviewer #3: Yes

3. Has the statistical analysis been performed appropriately and rigorously? 

Reviewer #1: Yes

Reviewer #3: I Don't Know

4. Have the authors made all data underlying the findings in their manuscript fully available?

Reviewer #1: Yes

Reviewer #3: Yes

5. Is the manuscript presented in an intelligible fashion and written in standard English?

Reviewer #1: Yes

Reviewer #3: Yes

6. Review Comments to the Author

Reviewer #1: (No Response)

Reviewer #3: The authors have addressed the reviewer’s comments. Therefore, I recommend accepting the manuscript.

7. PLOS authors have the option to publish the peer review history of their article (what does this mean?). If published, this will include your full peer review and any attached files.

Reviewer #1: No

Reviewer #3: No

---

## [Editor Report · Acceptance letter]

PONE-D-25-65983R1

PLOS One

Dear Dr. Rahaman,

I'm pleased to inform you that your manuscript has been deemed suitable for publication in PLOS One. Congratulations! Your manuscript is now being handed over to our production team.

Kind regards,

on behalf of

Dr. Latha Marasamy

Academic Editor

PLOS One